# Kondo breakdown in multi-orbital Anderson lattices induced by destructive hybridization interference

**Fabian Eickhoff[1]⋆ and Frithjof B. Anders[2]†**

**1** Institute For Software Technology, German Aerospace Center, 51147 Cologne, Germany
**2** Department of Physik, Condensed Matter Theory, TU Dortmund University,
44221 Dortmund, Germany

⋆ fabian.eickhoff@dlr.de , † frithjof.anders@tu-dortmund.de

## Abstract

In this paper we consider a multi-band extension to the periodic Anderson model. We use a single site DMFT(NRG) in order to study the impact of the conduction band mediated effective hopping of the correlated electrons between the correlated orbitals onto the heavy Fermi-liquid formation. Whereas the hybridization of a single impurity model with two distinct conduction bands always adds up constructively, $T_K \propto \exp(-\text{const}\, U/(\Gamma_1 + \Gamma_2))$, we show that this does not have to be the case in lattice models, where, in remarkable contrast, also an low-energy Fermi-liquid scale $T_0 \propto \exp(-\text{const}\, U/(\Gamma_1 - \Gamma_2))$ can emerge due to quantum interference effects in multi-band models, where $U$ denotes the local Coulomb matrix element of the correlated orbitals and $\Gamma_i$ the local hybridization strength of band $i$. At high symmetry points, heavy Fermi-liquid formation is suppressed which is associated with a breakdown of the Kondo effect. This results in an asymptotically scale-invariant (i.e., power-law) spectrum of the correlated orbitals $\propto |\omega|^{1/3}$, indicating non-Fermi liquid properties of the quantum critical point, and a small Fermi surface including only the light quasi-particles. This orbital selective Mott phase demonstrates the possibility of metallic local criticality within the general framework of ordinary single site DMFT.



# 1   Introduction

Condensed matter physics has long relied on the Fermi-liquid framework [1] to understand the low-temperature behavior of metals. However, certain correlated quantum materials defy this understanding, showcasing non-Fermi liquid behavior either as stable phases or through zero-temperature quantum phase transitions, captivating researchers exploring quantum criticality.

Initially research focused on magnetic to metallic transitions driven by spin-density-waves (antiferromagnetism or ferromagnetism) within the Landau-Ginzburg-Wilson (LGW) framework [2,3], but the spotlight now turns to even more peculiar transitions where LGW fails, notably observed in heavy Fermion materials [1,4]. This novel class of phase transitions cannot be understood in terms of local order parameter fluctuations of a collective magnetic or charge ordered phase.

Inelastic neutron scattering experiments have revealed valuable insights into these strange metallic quantum critical points (QCPs) [5–7]. These experiments detect unusual features in the frequency and temperature dependencies of dynamical spin susceptibility, marked by an anomalous power law exponent and low-frequency $\omega/T$ scaling. Significantly, this anomalous behavior is consistently observed throughout the Brillouin zone, indicating that these QCPs are governed by local critical fluctuations rather then collective excitations in the lattice.

In the context of heavy fermion materials, local criticality is often discussed as Kondo breakdown QCP (KB-QCP) [8–11], where localized and itinerant electrons effectively decouple at low energies leading to transition of a large to a small Fermi surface. This scenario is connected to orbital-selective Mott transitions (OSM) [12–14], where some electrons experience Mott localization while others remain itinerant. In these cases the emergence of magnetic order may occur as an instability of the Mott localized phase in order to remove the entropy of the local moments present in a paramagnetic Mott insulator, but is not necessarily required for the occurrence of the QCP.

While impurity problems with boundary phase transitions are by now well understood [15–18], the origin of local criticality in lattice problems still lacks a sufficient theoretical description.

Within the generic framework of single-site dynamical mean field theory (DMFT), it has been demonstrated, for instance, that the OSM phase is in general unstable against hybridization of the localized states with a Fermi surface [19]. This raises questions about the existence of the KB-QCP in the periodic Anderson model (PAM), which is commonly used to describe the fundamental properties of heavy fermion compounds and where such hybridization is always present.

In a single-site DMFT, the KB-QCP can only be stabilized if the hybridization is assumed to vanish precisely at the Fermi energy [20] or if two-dimensional magnetic bulk fluctuations are considered to effectively couple to the local moment degrees of freedom [4, 21], weakening the local nature of the QCP.

Alternatively, two-site cluster extensions of DMFT (CDMFT) have been applied to the PAM, revealing a KB-QCP stabilized by inter-site correlations [22–24]. This is a DMFT implementation of the Doniach picture [25] where an antiferromagnetic ordered phase mediated by RKKY interaction at small hybridization competes with heavy Fermi-liquid whose low energy scale is connected to the Kondo temperature. While the QCP is destroyed in the two-impurity model [26–28] due to particle-hole symmetry breaking it can be stabilized in a two-site CDMFT [24] due to the lattice self-consistency condition. Yet, within this extension, the strong inter-site correlations driving the phase transition are simultaneously neglected between sites of different clusters. It remains unclear whether the appearance of the KB-QCP is an artifact of this two-site cluster approximation or if it persists when the cluster size is systematically increased. A Kondo breakdown occurring in the magnetic phase of the half-filled KLM on the honeycomb lattice was reported by Raczkowski et al. [29]. In this scenario the question arrises how a realization of Dirac electrons leading to a Kondo breakdown transition in the impurity limit carries over to the half-filled lattice model.

In this paper, we propose a different mechanism promoting quantum criticality in heavy fermion materials, which solely relies on destructive quantum interference effects in a multi-band scenario, realized for example by depleted versions of the PAM.

While in single or few impurity problems the dynamic screening of the emerging local moments are governed by the Kondo effect, this does not hold for multi-impurity models [30–32] with large number of correlated orbitals. In such models the real part of the hybridization function becomes more important since there are not enough Kondo screening channels available. We demonstrate within a single-site DMFT, that in a multi-orbital extension of the common PAM, destructive interference in hybridization can lead to an exponentially reduced low-energy screening scale, the Kondo lattice temperature, without reducing the local Kondo coupling. This effect is unique to lattice models, as interference is always constructive in the corresponding single-impurity Anderson model (SIAM).

In addition to the fact, that such a reduction of the Kondo lattice temperature makes the system highly susceptible to the occurrence of magnetic order in the spirit of the traditional Doniach picture [25], which is beyond our single site DMFT analysis, we establish the existence of another critical point, which aligns with the concept of local criticality. Approaching these high symmetry points the Kondo lattice temperature can be continuously turned to zero, and the local moments gradually decouple from the Fermi surface. A line of local QCP emerge which are characterized by the degree of particle-hole asymmetry.

The paper proceeds as follows: Section 2 details the model and methodology. In Section 3, we provide a theoretical motivation and discuss recent developments in strongly correlated impurity models. Section 4 unveils our results, including the impact of destructive hybridization interference and the emergence of a Kondo breakdown quantum critical point. Finally we summarize and discuss our results in section 5.

## 2 Model and method

### 2.1 Model

In this study, we investigate a multi-orbital extension of the well-known periodic Anderson model (PAM). The model is defined as follows:

$$H = H^c + H^f + H^{hyb}. \tag{1}$$

The individual components of this Hamiltonian describe various aspects of the system: the non-interacting bands are denoted as $H^c$, the interacting flat band as $H^f$, and the hybridization term as $H^{hyb}$.

$$H^c = \sum_\nu^{N_\nu} \sum_{\vec{k},\sigma} (\epsilon^c_{\vec{k}\nu} + \epsilon^c_\nu) n^c_{\vec{k}\sigma\nu}, \tag{2}$$

$$H^f = \sum_l^{N_f} \epsilon^f (n^f_{l\uparrow} + n^f_{l\downarrow}) + U n^f_{l\uparrow} n^f_{l\downarrow}, \tag{3}$$

$$H^{hyb} = \sum_\nu^{N_\nu} \sum_{l,\sigma} V_\nu f^\dagger_{l\sigma} c_{l\sigma\nu} + \text{h.c.}, \tag{4}$$

with $n^c_{\vec{k}\sigma\nu} = c^\dagger_{\vec{k}\sigma\nu} c_{\vec{k}\sigma\nu}$ and $n^f_{l\sigma} = f^\dagger_{l\sigma} f_{l\sigma}$. The creation and annihilation operators $c^\dagger_{\vec{k}\sigma\nu}$ and $c_{\vec{k}\sigma\nu}$ create (annihilate) electrons with spin $\sigma$ and momentum $\vec{k}$ in the non-interacting band with band index $\nu$ and dispersion $\epsilon^c_{\vec{k}\nu} + \epsilon^c_\nu$, while $c^\dagger_{l\sigma}$ ($c_{l\sigma}$) and $f^\dagger_{l\sigma}$ ($f_{l\sigma}$) create (annihilate) an electron with spin $\sigma$ in the $c/f$-orbital at the real space lattice site $l$. $N_\nu$ denotes the number of conduction bands and $N_f$ the number of correlated f-sites. The correlation effect of the $f$-orbitals is taken into account by the local Coulomb interaction of strength $U$. It is important to note that due to the hybridization $V_\nu$, the particle number in each band $\nu$ is not a conserved quantity, distinguishing it from multi-channel models with conserved channel symmetry [33–35].

In most studies of these kind of models only a single dispersive band $\nu$ is considered, however, as we explain in detail in Sec. 4.2, multiple bands inherently emerge in depleted versions of the standard PAM for example. If the lattice site index $l$ in Eqs. (3) and (4) only exhausts a subset of $N_f$ different lattice sites, the model is known as multi-impurity Anderson model (MIAM), where the special case $N_f = 1$ corresponds to the well understood SIAM. If, on the other hand, $l$ exhausts all lattice sites we restore the translational invariant PAM [36, 37].

### 2.2 Method

To tackle this complex model in the translational invariant form (PAM), we employ the DMFT [38, 39], which maps the lattice problem onto an effective single impurity Anderson model. The central DMFT equation, where we have dropped the spin index $\sigma$ for simplicity, reads:

$$G^f_{\text{lat}}(z) \overset{!}{=} G^f_{\text{imp}}(z). \tag{5}$$

Here, $G^f_{\text{lat}}(z)$ and $G^f_{\text{imp}}(z)$ describe the local single-particle $f$-Greens function of the lattice and the effective impurity problem, respectively. These spectral functions are formally defined as:

$$G^f_{\text{lat}}(z) = \sum_{\vec{k}} \left( z - \epsilon^f - \Delta^{\text{lat}}_{\vec{k}}(z) - \Sigma(z) \right)^{-1}, \tag{6}$$

$$G^f_{\text{imp}}(z) = \left( z - \epsilon^f - \Delta^{\text{eff}}(z) - \Sigma(z) \right)^{-1}. \tag{7}$$

In these equations $\Sigma(z)$ is the correlation-induced part of the self-energy, and $\Delta^{\text{eff}}(z)$ characterizes the effective medium in which the single impurity is embedded. The momentum dependent hybridization function $\Delta_{\vec{k}}^{\text{lat}}(z)$ is the physically relevant property, containing the information about the lattice model under investigation:

$$\Delta_{\vec{k}}^{\text{lat}}(z) = \sum_{\nu} V_{\nu}^2 G_{\vec{k}\nu}^{0c}(z), \tag{8}$$

where $G_{\vec{k}\nu}^{0c}(z) = (z - \epsilon_{\vec{k}\nu}^{c} - \epsilon_{\nu}^{c})^{-1}$ represents the propagator of a particle in the $\nu$-th non-interacting band. From Eq. (5) one can formulate the DMFT self-consistency equation as

$$\Delta^{\text{eff}}(z) = z - \epsilon^{f} - \Sigma(z) - (G_{\text{lat}}^{f}(z))^{-1}. \tag{9}$$

Importantly, $\Sigma(z)$ itself depends on the effective medium $\Delta^{\text{eff}}(z)$, making Eq. (9) a self-consistent problem. The only approximation made by DMFT is to neglect the momentum dependence of the self energy, $\Sigma_{\vec{k}}(z) \approx \Sigma(z)$, which gets exact for a lattice with infinite coordination number, i.e. in infinite dimensions [38, 39]. In order to get a self-consistent solution we proceed as follows:

(I)    initialize the effective Medium: $\Delta^{\text{eff}}(z) = \sum_{\vec{k}} \Delta_{\vec{k}}^{\text{lat}}(z)$,

(II)    solve the impurity problem defined by $\Delta^{\text{eff}}(z)$ to obtain the self-energy $\Sigma(z)$,

(III)    calculate $G_{\text{lat}}^{f}(z)$ from Eq. (6) (see appendix A and Eq. (A.1) respectively),

(IV)    calculate $\Delta^{\text{eff}}(z)$ from Eq. (9),

(V)    continue steps (II) to (IV) until self-consistency is reached.

It is important to note, that while there always exists an equivalent single-band SIAM in case of a multi-band SIAM, this is generally not possible in the context of a multi-band extension to the PAM as discussed here. For more details on this topic, please refer to appendix B.

## 3    Theoretical motivation

In this section, we delve into the theoretical motivation behind our study, focusing on the interplay between dispersive bands and correlated $f$-orbitals in lattice models. We begin by considering a specific scenario with a single dispersive band, $N_{\nu} = 1$, described by Eq. (2) and $N_f$ correlated $f$-orbitals coupled to selected lattice sites.

The influence of the band onto the dynamics of these $f$-orbitals is completely determined by the complex hybridization function, denoted as:

$$\Delta_{lm}(z) = V^2 G_{lm}^{0c}(z), \tag{10}$$

where $G_{lm}^{0c}(z)$ describes the propagation of an electron from site $m$ to site $l$ in the non interacting band.

In a previous study [30], we conducted a thorough analysis of the hybridization matrix in Eq. (10), unveiling intriguing insights. We observed that as the number of correlated orbitals, $N_f$, increases, the relevance of the imaginary part diminishes, while the significance of the real part grows. The complex hybridization matrix $\Delta_{lm}(\omega + i\delta)$ can be split into a real and imaginary part for $\delta \to 0^+$:

$$\begin{aligned}\Delta_{lm}(\omega + i\delta) &= \Re\Delta_{lm}(\omega) + i\Im\Delta_{lm}(\omega) \\ &= T_{lm}(\omega) + i\Gamma_{lm}(\omega).\end{aligned} \tag{11}$$

We have shown that the dynamics of a MIAM with $N_f$ sites is determined by the matrices at $\omega = 0$ in the wide band limit. While $T_{lm}(0)$ is interpreted as an effective hopping between two correlated sites $l$ and $m$, the eigenvalues of $\Gamma_{lm}$ determine the coupling to the effective conduction bands. The number of such bands is given by the rank of the matrix $\Gamma_{lm}$. While the rank is $N_f$ for small values of $N_f$, we have proven that for $N_f$ exceeding a certain threshold which dependent on model specifics that $\text{rank}(\Gamma_{lm}) < N_f$. This implies that for large $N_f$, many eigenvalues of the matrix $\Gamma_{lm}$ must be zero, the notion of individual Kondo screening via an effective conduction band coupling does not make sense any longer. $T_{lm}(0)$ becomes dominant and neglecting the real part will even alter the nature of the low-energy fixed point (FP) since it is responsible for the Fermi-liquid formation in a similar way as observed in the Hubbard model. In fact, for $\text{rank}(\Gamma_{lm}) = 0$, the model becomes a Hubbard model in the wide band limit with a single particle inter-site hopping $T_{lm}(0)$, as demonstrated in Sec. II.F.3 of Ref. [30]. For an in-depth exploration of this analysis, refer to Fig. 1 in Ref. [30] and the accompanying explanations.

Our understanding of these hybridization function intricacies has enabled the description of local moment formation in Kondo-Hole systems [31] and the evolution of the hybridization gap in the spectral function of strongly correlated impurity clusters [32]. These findings are in notable agreement with rigorous theoretical statements made by the Lieb-Mattis theorem [40, 41].

In this study, we shift our focus to another aspect, prompted by the aforementioned crossover behavior, which holds particular relevance for lattice models with more than one $f$- and one $c$-orbital per unit cell. In such scenarios, the hybridization function in Eq. (10) undergoes slight modifications and can be expressed as:

$$\Delta_{lm}(z) = \sum_\nu V_\nu^2 G_{\nu,l,m}^{0c}(z). \tag{12}$$

Of note, the sign of the imaginary part of individual propagators remains fixed due to causality, ensuring that their contributions consistently sum up. In the case of the SIAM, this implies that the effective hybridization with the continuum can only increase, resulting in an enhanced Kondo temperature.

Conversely, the sign of the real part remains undetermined, allowing contributions from different bands to potentially offset or even cancel each other out. Consequently, if the screening of the local moment is primarily dominated by the real part of the propagator, additional bands can have strong influence and might even suppress the low-energy screening scale.

In the subsequent sections, we employ the combination of DMFT and the numerical renormalization group (NRG) approach to test this hypothesis within the framework of the PAM described by Eq. (1), while considering the presence of two conduction bands.

# 4 Results

To address the properties of the effective SIAM arising from the DMFT self-consistency equation, Eq. (5), we employ the Numerical Renormalization Group (NRG) technique, as implemented in the NRG-Lubljana interface [42, 43] to the TRIQS open-source package [44]. Details on how we accurately solved the DMFT integrals can be found in appendix A.

Unless explicitly stated otherwise we use a constant density of state (DOS) for the two dispersive bands, $N_\nu = 2$ and $\rho^c(\omega) = \Theta(|\omega| - D_\nu)\frac{1}{2D_\nu}$, where $D_1 = D_2 = D$ represent the individual band widths, which are chosen to be identical. This ensures consistency and avoids complications in the interpretation of the results arising from frequency-dependent DOS.

In addition, the centers of the individual bands, $\epsilon_\nu^c$, are always shifted from the Fermi energy, $|\epsilon_\nu^c| > 0$, in order to avoid the Kondo insulator regime when recovering the limit of the standard PAM with only one conduction band [36].

It is important to emphasize that our results maintain their qualitative validity regardless of the specific assumptions made, including the choice of the DOS and the uniform band widths.

In order to analyze the influence of the second band onto the screening of the local moment at each correlated site $l$ without changing the overall strength of hybridization, we define a parameter $\alpha$

$$V_2^2 = \alpha V_1^2 = \frac{\alpha}{1+\alpha} V^2, \tag{13}$$

such that $V_1^2 + V_2^2 = V^2$ is always fulfilled. Consequently, we can use $\Gamma$ as our unit of energy, which is defined as:

$$\Gamma = \Gamma_1 + \Gamma_2 = \frac{\pi V^2}{2D}. \tag{14}$$

Notably, by choosing the definition in Eq. (13) we ensure independence of the leading order of the Kondo temperature of the corresponding SIAM with respect to $\alpha$ [45]:

$$T_K \propto \sqrt{\Gamma U} \exp\left(-\frac{\pi U}{8\Gamma}\right), \tag{15}$$

for a particle-hole symmetric f-site, $\epsilon^f = -U/2$.

This consistent framework allows for meaningful comparisons and facilitates a clear understanding of the physical behavior in multi-orbital Anderson lattices.

## 4.1 Destructive hybridization interference

Within the framework of DMFT, the low energy scale in the lattice problem, denoted as $T_0$, shares a similar analytical form to Eq. (15), as we effectively map the model onto an effective SIAM. However, due to the self-consistency procedure, the effective hybridization, represented as $\tilde{\Gamma}$, can undergo significant renormalization. To gain insight into this renormalization within the lattice context, we systematically investigate the influence of a second band.

As evident from the DMFT equation (5), the influence of the bands on the dynamics of the $f$-orbitals is solely determined by the weighted sum of individual single-particle Green's functions of the band electrons,

$$\sum_\nu V_\nu^2 G_{\vec{k}\nu}^{0c}(z) = \frac{V_1^2}{z - (\epsilon_{\vec{k}1}^c + \epsilon_1^c)} + \frac{V_2^2}{z - (\epsilon_{\vec{k}2}^c + \epsilon_2^c)}, \tag{16}$$

where its analytic low frequency properties, $\omega/D \ll 1$, significantly determines the low-temperature physics of the problem.

To meticulously investigate the impact of destructive hybridization interference, we now make the assumption that the dispersion relations of the two bands satisfy the following relation:

$$\epsilon_{\vec{k}1}^c + \epsilon_1^c = -(\epsilon_{\vec{k}2}^c + \epsilon_2^c). \tag{17}$$

While the choice of the dispersion for the second band, mirroring the first one about the Fermi energy, might seem arbitrary at this point, we will clarify in the next subsection (4.2) that such a behavior can indeed be achieved through a remarkably straightforward real-space geometry.

Inserting the equations (13) and (17) into Eq. (16), and focusing on the zero frequency contribution, $\omega = 0$, we obtain

$$\sum_\nu V_\nu^2 G_{\vec{k}\nu}^{0c}(0) = \frac{1-\alpha}{1+\alpha} \frac{-V^2}{\epsilon_{\vec{k}1}^c + \epsilon_1^c} - i\pi V^2 \delta(\epsilon_{\vec{k}1}^c + \epsilon_1^c), \tag{18}$$

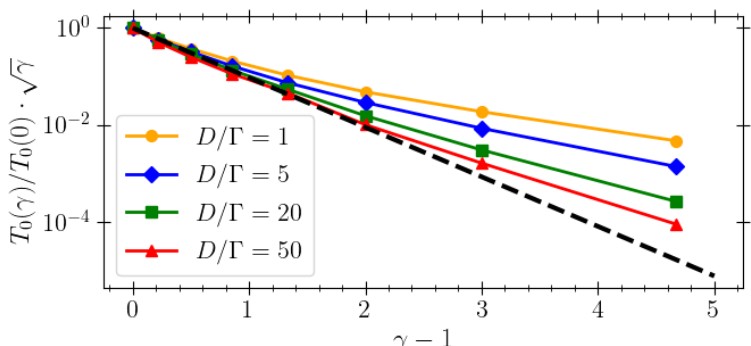

Figure 1: Low energy scale $T_0$ as function of the dimensionless parameter $\gamma(\alpha) = \frac{1+\alpha}{1-\alpha}$ for different band widths $D$, $U/\Gamma = 6$ and $\epsilon_1^c/D = 0.6$. The dashed black line corresponds to the function $f(\gamma) = \exp(-\frac{\pi U}{8\Gamma}(\gamma - 1))$.

where the contributions are separated into real and imaginary components.

Remarkably, the imaginary part remains independent of $\alpha$, while the real part features a prefactor of $\frac{1-\alpha}{1+\alpha}$. This observation makes this model perfectly suited to test our hypothesis that in contrary to a SIAM where the imaginary part of the hybridization function determines the influence of the conduction band onto local dynamics, the real part becomes the dominant property in a lattice system where $\mathrm{rank}(\Gamma_{lm}) < N_f$. In a multi-band setup, the physics of the PAM can change due to destructive interference at the Fermi energy.

In the following, we focus on the paramagnetic solution and enforce a spin degenerate self-energy, $\Sigma_\sigma(z) = \Sigma(z)$, within the DMFT cycle by assuming conservation of total spin quantum number in NRG. We adhere to Wilson's definition [46], and define the lattice low energy scale $T_0$ [47] as the temperature at which the spin susceptibility of the impurity embedded in the converged effective medium reaches the value of $(g\mu_B)^2 \chi_{\mathrm{imp}}(T_0) = 0.07$.

Figure 1 displays the low energy scale $T_0(\gamma)/T_0(\gamma = 1)\sqrt{\gamma}$ vs the dimensionless parameter $\gamma(\alpha) = \frac{1+\alpha}{1-\alpha}$ for various band widths, $U/\Gamma = 6$ and $|\epsilon_\nu^c|/D = 0.6$. The plots clearly demonstrate the exponential dependence of the low energy scale $T_0$ on $\gamma$. This corroborate our conjecture that the screening of the local moment in a periodic model is mainly driven by the real part of the conduction band propagator and is not dominated by the imaginary part. This surprising and somehow counterintuitive result is in contrast of the observation in the SIAM and the two-impurity Anderson model [48, 49].

By approximating the propagator at zero frequency ($\omega/D \approx 0$) in Eq. (18), we can determine the renormalization of the hybridization from its real part: $V^2(\gamma) = V^2/\gamma$. If we substitute the renormalized hybridization into Eq. (15) for the single impurity Kondo temperature, the ratio of the low temperature scales should follow the relation

$$\sqrt{\gamma}\,\frac{T_0(\gamma)}{T_0(\gamma = 1)} = \exp\left(-\frac{\pi U}{8\Gamma}(\gamma - 1)\right). \tag{19}$$

Note that $\gamma(\alpha = 0) = 1$ corresponds to the conventional single band PAM, and $\gamma - 1 = 2\alpha/(1-\alpha)$ is a measure how much the second band contributes to the local dynamics.

The r.h.s of Eq. (19) is added as a dashed black line to Figure 1. For small values of $\gamma - 1$, we observe an qualitatively good agreement between the ratio determined numerically with the DMFT(NRG) and analytic expression stated in Eq. (19). As $\gamma$ increases, the zero frequency estimate $V^2(\gamma) = V^2/\gamma$ tends to overestimate the renormalization. However, for larger band widths, the deviation from Eq. (19) becomes smaller. This behavior can be attributed to the fact that Eq. (16) is strictly valid at $\omega/D \approx 0$, and the frequency dependence of the propa-

gator becomes more influential for small band widths $D$. For smaller band width, the finite frequencies start to contribute stronger which leads to deviations from the analytic estimate.

By tuning the parameter $\alpha$ from 0 to 1, we continuously increase the admixture of the second conduction band. $\alpha = 0$ corresponds to the conventional single band PAM away from the Kondo insulator case, and $\alpha = 1$ to a two-band model with perfect destructive interference of the real part of the conduction band propagators. We demonstrated in Fig. 1 that the low temperature scale of the lattice, $T_0$, which has the meaning of an effective Kondo lattice scale, vanishes for $\alpha \to 1$ ($\gamma(\alpha \to 1) \to \infty$). The model undergoes a quantum phase transition: At the coupling $\alpha = 1$, the Kondo screening breaks down and the effective impurity has a stable local moment (LM) FP.

Please note that we leveraged a highly specific assumption regarding the second band, which mirrors the characteristics of the first band around the Fermi energy. This approach was chosen for its ability to provide a straightforward and lucid demonstration of the existence and significance of destructive hybridization interference. In essence, this phenomenon manifests whenever the absolute value of the sum of the real parts of individual contributions is smaller than the sum of the absolute values of these contributions:

$$\left| \sum_\nu V_\nu^2 \Re G_{\vec{k}\nu}^{0c}(0) \right| < \sum_\nu \left| V_\nu^2 \Re G_{\vec{k}\nu}^{0c}(0) \right| . \tag{20}$$

This condition doesn't necessitate bands to be precisely inverse, rather, they should exhibit distinct curvatures around the Fermi energy. This broader context resembles our scenario where $V_1 \neq V_2$ such that $\alpha \neq 1$.

In summary, the findings impressively demonstrate the potential for destructive hybridization interference within the PAM. Note, that in our DMFT(NRG) calculation, we use the full energy dependency of the medium $\Delta_\sigma(z)$, determined by the Eqs. (5)-(7). We used the fundamental findings of our MIAM study [30–32] by focusing on the value of $\Delta_\sigma(z)$ at the Fermi energy, only to construct the model and predict the outcome of the full DMFT. This demonstrates again the powerful concept that in such problems with infrared divergence physics the low energy excitations play the crucial role. Since the parameter regimes are adiabatically connected deviations from the wide-band limit does not alter the physics but only lead to some perturbative corrections to the simple analytical predictions as seen in Fig. 1.

## 4.2 Generality of destructive hybridization interference

In the previous subsection 4.1 we worked out the main result of this paper: If a flat band of interacting electrons hybridizes with more than one conduction band, the low temperature screening scale $T_0$ (Kondo lattice temperature) can be exponentially reduced due to destructive interference reducing the effective f-f hopping. However, to achieve this effect, the dispersions of the conduction bands need to have a different curvature such that quantum interference in Eq. (16) is destructive at low frequencies. Therefore, with Eq. (17), we made the somewhat unconventional assumption that the two bands are precisely mirrored. In this section we are going to demonstrate that such destructive interference indeed arises if more general multiorbital versions of the standard PAM are considered.

Perhaps the simplest way of extending the original PAM, which has only one conduction band ($N_\nu = 1$), into a multi-orbital model is by regularly removing some of the $f$-orbitals such that (a) lattice periodicity is maintained, (b) the lattice spacing of the $f$-lattice is enlarged compared to the $c$-lattice and (c) the extended unit cell still contains exactly one single $f$-orbital. This introduces multiple bands as we need to fold the original dispersion of the $c$-electrons into the reduced Brillouin zone of the larger unit cell.

By switching off the coupling $V$ between the $c$- and $f$-subsystem, we can use the usual Fourier transformation into momentum space for both subsystems individually to diagonalize

the bilinear part. However, due to the different lattice spacing of the $c$- and $f$-lattice, the related reduced Brillouin zones $\mathrm{Bz}^c$ and $\mathrm{Bz}^f$ differ as well. If we suppress the spin index $\sigma$ for simplicity, the hybridization part reads:

$$H^{\mathrm{hyb}} = V \sum_l f_l^\dagger c_l + \mathrm{h.c.} \tag{21}$$

$$= \frac{V}{\sqrt{N_f N_c}} \sum_{\vec{k} \in \mathrm{Bz}^c} \sum_{\vec{q} \in \mathrm{Bz}^f} \sum_l e^{i(\vec{k}-\vec{q})\vec{R}_l} f_{\vec{q}}^\dagger c_{\vec{k}} + \mathrm{h.c.} \tag{22}$$

$$= \frac{V}{\sqrt{N_c^u}} \sum_{n=1}^{N_c^u} \sum_{\vec{q} \in \mathrm{Bz}^f} f_{\vec{q}}^\dagger c_{\vec{q}+\vec{p}_n} + \mathrm{h.c.}, \tag{23}$$

where $N_c^u = N_c/N_f$ denotes the number of $c$-orbitals within the enlarged unit cell. We used the relation

$$\sum_l e^{i(\vec{k}-\vec{q})\vec{R}_l} = N_f \sum_n \delta_{\vec{k},\vec{q}+\vec{p}_n}, \tag{24}$$

where the set of momentum vectors $\vec{p}_n$ depends on the basis vectors $\vec{\delta}_i$ of the $f$-lattice, $\vec{R}_l = \sum_i \alpha_{il} \vec{\delta}_i$, and is determined by the two conditions:

$$\vec{p}_n \in \mathrm{Bz}^c \quad \wedge \quad \vec{p}_n \vec{\delta}_i = 2\pi n; \, n \in \mathbb{N}. \tag{25}$$

This relates $\vec{k} \in \mathrm{Bz}^c$ to $\vec{q} \in \mathrm{Bz}^f \subset \mathrm{Bz}^c$ where $\vec{p}_n$ are the momentum vectors for backfolding the c-band band structure into the Brillouin zone $\mathrm{Bz}^f$.

As a result, the propagator of the $f$-electrons becomes:

$$G_{\vec{q}\sigma}^f(z) = \left(z - \epsilon^f - \tilde{G}_{\vec{q}}^c(z) - \Sigma_{\vec{q}\sigma}(z)\right)^{-1}, \tag{26}$$

$$\tilde{G}_{\vec{q}}^c(z) = \frac{1}{N_c^u} \sum_n V^2 G_{\vec{q}+\vec{p}_n}^{0c}(z). \tag{27}$$

$G_{\vec{q}+\vec{p}_n}^{0c}(z) = (z - \epsilon_{\vec{q}+\vec{p}_n})^{-1}$ describes the propagation of an electron in the respective piece $n$ of the original conduction band, $\epsilon_{\vec{k}}$ with $\vec{k} = \vec{q} + \vec{p}_n \in \mathrm{Bz}^c$, folded into $\mathrm{Bz}^f$.

Lets quickly discuss three special cases: (I) the standard PAM, (II) the SIAM and (III) minimal depletion on a cubic lattice:

I. In this case the first Brillouin zones of the $f$- and $c$- lattice are identical, $N_c^u = 1$ and $\vec{\delta}_l$ describes the basis vectors of both lattices. The only solutions to Eq. (25) are the reciprocal lattice vectors $\vec{g}_l$, for which $G_{\vec{q}+\vec{g}_l}^{0c}(z) = G_{\vec{q}}^{0c}(z)$ holds, and, consequently, Eq. (27) reduces to the well known form for the PAM $\tilde{G}_{\vec{q}}^c(z) = V^2 G_{\vec{q}}^{0c}(z)$.

II. By increasing the distance $|\vec{\delta}_l|$ between the $f$-orbitals we can reach the limit of the SIAM for $|\vec{\delta}_l| \to \infty$. In this limiting case the first Brillouin zone of the $f$-orbitals contains only a single point (which means that we can drop the index $\vec{q}$), while Eq. (25) is fulfilled by every point $\vec{k}$ within the first Brillouin zone of the $c$-lattice. Therefore, Eq. (27) becomes $\tilde{G}_{\vec{q}}^c(z) \to \tilde{G}^c(z) = \sum_{\vec{k}} V^2 G_{\vec{k}}^{0c}(z)$, which indeed is the hybridization function of the SIAM.

III. Since the cubic lattice is bipartite, the minimal periodic depletion on such a geometry corresponds to removing the $f$-orbitals of one of the sublattices. Even if this is only one specific realization of an arbitrary depletion pattern, this model is commonly referred to as the "depleted PAM" [50–52]. For simplicity we will focus on the one dimensional case, however, the result does not change in higher dimensions.

The unit cell of the $f$-lattice is twice as large as the one of the $c$-lattice: $\delta = 2a$, and, consequently, Eq. (25) has two solutions: $p_0 = 2\pi/a$ and $p_1 = \pi/a$.

Assuming $\epsilon_k = D\cos(ka)$ for the $c$-band we obtain $\epsilon_{q+p_0} = \epsilon_q$ and $\epsilon_{q+p_1} = -\epsilon_q$, such that Eq. (27) reads:

$$\tilde{G}^c_q(z) = \frac{V^2}{2}\left(\frac{1}{z - \epsilon^c_q} + \frac{1}{z + \epsilon^c_q}\right). \tag{28}$$

The structure of Eq. (28) is identical to the one in Eq. (16), describing the hybridization with two dispersive bands. In addition, the sign in front of the dispersive part of Eq. (27) is different for these bands, matching our assumption in Eq. (17).

The physics of the depleted PAM [50–52] at half-filling is well understood. In accordance with the extended Lieb-Mattis theory [41] the ground state has a ferromagnetic polarisation of $S^{\text{tot}}_Z = (N_f - 1)/2$ since the Kondo effect only screens a single spin of the lattice. We have shown [30] that the rank of the hybridization function is $\text{rank}(\Gamma_{im}) = 1$, and, therefore, only a single Kondo screening channel is available which is irrelevant in the continuum limit. In the paramagnetic phase, spin ordering is not considered and free local moments are found within a DMFT approximation that ignores inter-site two-particle correlations. This physics corresponds to $\alpha = 1$ in the previous section, where we have explicitly shown that the low-energy scale vanishes in this specific form of the PAM treated with the DMFT.

In summary, Eq. (26) shows, that a regularly depleted version of the PAM with only a single conduction band $\epsilon_{\vec{k}}$ is equivalent to a model described by Eq. (1). The different bands $\epsilon_{\vec{k}\nu}$ correspond to different pieces of the original conduction band $\epsilon_{\vec{k}}$, which emerge by folding it into the reduced Brillouin zone of the $f$-lattice. Of course these pieces will have different curvatures in general, such that destructive interference within the hybridization can lead to a strongly reduced Kondo lattice temperature $T_0$.

In Sec. 4.1 we used a concerted two-band model as a simplified example to study the effect of destructive interference in a systematic way, by tuning the parameter $\alpha$. However, depletion of $f$-orbitals in the PAM has a similar effect, leading to a modified $\alpha$: $0 \leq \alpha \leq 1$. Consequently, destructive hybridization interference is a recurring phenomenon in multi-orbital extensions of the PAM and will hold a pivotal role in elucidating the intriguing characteristics of heavy Fermion materials, where the low energy physics is governed by $T_0$.

## 4.3 Spectral functions and Kondo breakdown QCP

Having established the potential for destructive hybridization interference in the multi-orbital extension of the PAM, we will use our auxiliary two-band model to focus on the special scenario characterized by symmetric couplings, $\alpha = V_2/V_1 = 1$, where Eq. (19) predicts a complete suppression of the low-energy screening scale. We systematically explored various parameter sets, encompassing diverse shapes for the $c$-orbital density of states (DOS), the ratio of $D_1/D_2$, and particle-hole asymmetric $f$-orbitals.

In the strongly correlated limit, where local moments develop at intermediate temperature, we could not identify any combination of these parameters that would yield a solution with a finite $T_0 > 0$. In stark contrast, the converged DMFT solution for the SIAM consistently resides in the LM FP. This confirms our initial conjecture that the screening of the local moments on the f-sites is connected to the effective inter f-site hopping mediated by the effective hopping matrix elements $T_{lm}(0)$ in a PAM and not by initial local hybridization $\Gamma_{ll}(0)$ which is only relevant for MIAMs with the $\text{rank}(\Gamma_{lm}) = N_f$ which excludes the PAM. We would like to emphasize that this effect is included in the DMFT self-consistency equations, and our analysis only reveals its underlying mechanism.

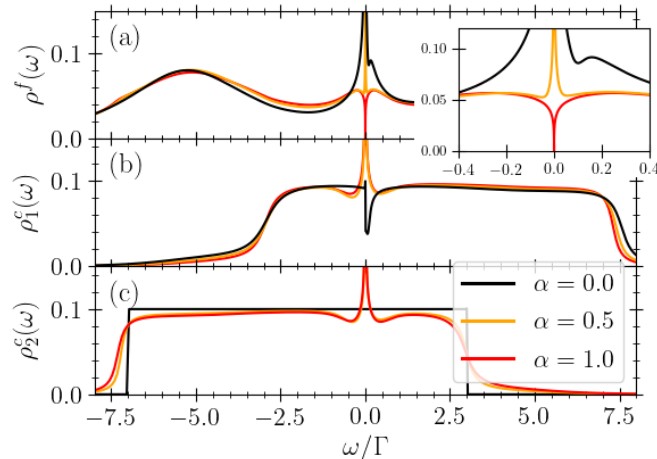

Figure 2: Single-particle spectral function of (a) the $f$-orbitals, $\rho^f(\omega)$ (b) the $c_1$-orbitals, $\rho^c_1(\omega)$ and (c) the $c_2$-orbitals, $\rho^c_2(\omega)$, of the paramagnetic DMFT solution. The black line represents the case where only the $c_1$ orbitals hybridize with the $f$-orbitals ($\alpha = 0.0$), while the red line depicts results for identical hybridization with both bands ($\alpha = 1.0$). The inset displays the $f$-spectra for small frequencies. The model parameters used are: $D/\Gamma = 5$, $U/\Gamma = 10$, and $|\epsilon^c_\nu|/D = 0.4$.

We split this subsection into two parts. In the first part, we focus on the special case of locally particle-hole (PH) symmetric $f$-orbitals, whereas we allow particle hole asymmetry in the second part.

### 4.3.1 Particle-hole symmetric $f$-orbitals

Here we concentrate on the special case of PH symmetric $f$-orbitals by setting $\epsilon^f = -U/2$. Note however, that due to $|\epsilon^c_\nu| \neq 0$, the Hamiltonian remains PH asymmetric as long as $\alpha \neq 1$. PH-symmetry is achieved for $\alpha = 1$.

A word is in order to point out the differences to the single band PH symmetric model, $\alpha = 0, |\epsilon^c_\nu| = 0$ which is a Kondo insulator. In the latter case, $T_{lm}(0) \neq 0$ and the effective $f$-site does not only couple to the conduction band media $\tilde{\Gamma}(\omega)$ but also to an effective free orbital that has $f$-orbital character as pointed out by Pruschke et al. [36]. Within the DMFT, this coupling shows up as a $\delta$-peak at $\omega = 0$ in $\tilde{\Gamma}(\omega)$ and is essentially generated by the finite $T_{lm}(0)$ to the neighboring $f$-sites. Consequently, a local RKKY type singlet emerges despite of the gapped $\tilde{\Gamma}(\omega)$, which would normally lead to a LM FP. Essentially, Pruschke et al. [36] divided $\tilde{\Gamma}(\omega)$ into two parts: one stemming for the conduction electrons which shows a pseudo-gap and, therefore, does not provide a screening channel, and a $\delta$-peak stemming from the neighboring f-site which is being responsible for the singlet formation. This is in perfect alignment with the MIAM theory [30].

We present the spectral functions of the $f$-orbitals in Fig. 2(a), the $c_1$-orbitals in Fig. 2(b), and the $c_2$-orbitals in Fig. 2(c) for three different values of $\alpha$, $\alpha = 0.0$ (black line), $\alpha = 0.5$ (orange line), $\alpha = 1.0$ (red line), and the parameters: $D/\Gamma = 5$, $U/\Gamma = 10$, and $|\epsilon^c_\nu|/D = 0.4$.

We begin with the results for $\alpha = 0.0$ (black lines), where only the $c_1$-orbitals hybridize with the correlated $f$-orbitals ($V_2 = 0$), and we recover the standard PAM. Since the $c_2$-orbitals are entirely decoupled, their spectral function is identical to the initially assumed flat DOS with a width of $D$: $\rho^c_2(\omega) = \Theta(|\omega| - D)\frac{1}{2D}$.

The spectra of the $f$-orbitals exhibit the well-known Hubbard bands around $\omega/\Gamma \approx \pm U/2$ due to the particle-hole symmetric choice of $\epsilon^{\mathrm{f}} = -U/2$. Additionally, the Kondo resonance with a width of approximately $T_0$ around the Fermi energy ($\omega = 0$) is clearly visible. The dip near the Fermi energy, present in both the $f$ and $c_1$-orbital spectra, resembles the nature of hybridized bands. This dip evolves into a complete gap at $\omega = 0$ in the limit of the Kondo insulator regime for $\epsilon_1^{\mathrm{c}} \to 0$ [36].

Having reviewed the spectral functions in the context of the standard PAM with a single conduction band, we now shift our focus to $\alpha = 1$ (red lines).

While the Hubbard bands in the $f$-spectra remain largely unchanged, the Kondo resonance has collapsed. This suppression of the Kondo resonance aligns with the prediction from Eq. (19), which anticipates the vanishing of $T_0$ when $\gamma(\alpha = 1) = \infty$. Indeed, as $\alpha$ increases from 0, the Kondo resonance rapidly (exponentially) narrows, exemplarily shown for $\alpha = 0.5$ (orange lines).

In stark contrast, the $c$-orbitals manifest a divergence right at the Fermi energy, indicative of an overall metallic solution such that the Kondo breakdown falls within the class of orbital selective Mott transitions (OSM) [12–14]: The half-filled f-bands become an Mott insulator which is characterized by one local moment pre correlated site in the paramagnetic phase. Note that due to the RG flow, these moments are typically extended and not localized at the f-sites [31, 53]. In essence the f-electrons are excluded from the Fermi-surface. Including the remaining two-particle interactions between the f-spins which is beyond the single-particle DMFT, we expect a magnetic ordering of the $f$-momenta while the metallic properties are coming from the light quasiparticles.

### 4.3.2  Properties of the QCP for PH symmetric f-sites

After establishing that the critical coupling $\alpha = 1$ belongs to a quantum critical point which separates two strong coupling Fermi-liquid FPs we investigate the power law in the spectral functions right at the KB-QCP. We employed three different shapes for the original $c$-orbital DOS: constant, semicircular (Bethe lattice), and Gaussian (hypercubic lattice). In case of non constant DOS (semicircular and Gaussian) we replace Eq. (14) by $\Gamma = \pi V^2 \rho(\epsilon^{\mathrm{c}})$. Furthermore, we explore various sets of model parameters encompassing $D_\nu/\Gamma \in \{1, 5, 10\}$, $\epsilon^{\mathrm{c}}/D \in \{0.2, 0.7\}$, and $U/\Gamma \in \{4, 6, 10\}$.

Figure 3 illustrates the positive part of the different spectra on double logarithmic plots. Figure 3(a) displays the $c_1$-orbital spectra and Fig. 3(b) the $f$-orbital spectra. For better comparison, we plotted $\rho(\omega)/\rho(\omega_0)$, with $\omega_0/\Gamma = 10^{-5}$. As expected for the low-temperature physics of a QCP the spectra at high energies significantly differ for the different parameter sets but a universal low energy regime emerges. This low-frequency behavior adheres to a power law relationship, denoted as $\propto |\omega|^r$. We obtain the approximate exponents of $r^{\mathrm{f}} \approx 0.33$ for the $f$-spectra and $r^{\mathrm{c}} \approx -0.33$ for both $c$-spectra. Remarkably, this exponent remains unaffected by variations in the model parameters, emphasizing the concept of universality.

It's crucial to emphasize that the KB-QCP remains unaffected by the strength of the Coulomb interaction, effectively rendering $U > U_c = 0$. Indeed, this phase transition is discernible even in the band structure of the non-interacting model ($U = U_c = 0$). Given three orbitals per unit cell, the band structure comprises three bands of specific widths that depend on model parameters. In the case of PH-symmetric $f$-orbitals, $\epsilon^f = U = 0$, and $\alpha \to 1$, one of these bandwidths is gradually suppressed, resulting in a completely flat band precisely at the Fermi energy when $\alpha = 1$. This observation isn't unexpected, as the local moment FP of the SIAM is always continuously connected to the trivially decoupled FP for $V = 0$. Thus, any transition from strong coupling to the local moment FP within single-site DMFT can be delineated by a non-interacting Gaussian FP.

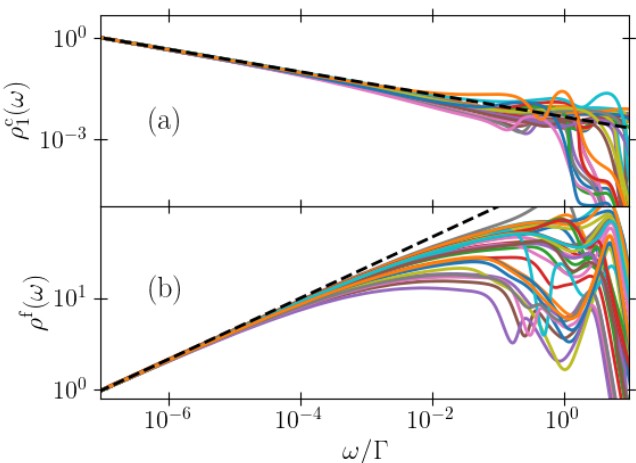

Figure 3: Single-particle spectral function of the (a) $c_1$-orbitals and (b) $f$-orbitals on a double logarithmic scale for different sets of parameters. The spectra are normalized to match at $\omega/\Gamma = 10^{-5}$. We employed three different DOS shapes: constant, semi-circular, and Gaussian. These were combined with varying values of $D/\Gamma \in \{1, 5, 10\}$, $\epsilon^c/D \in \{0.2, 0.7\}$, and $U/\Gamma \in \{4, 6, 10\}$. The dashed black lines represent the power law relationships: (a) $f(\omega) \propto |\omega|^{-0.33}$, (b) $f(\omega) \propto |\omega|^{0.33}$.

When we introduce a finite bandwidth for the $f$-orbitals, the critical value $U_c$ shifts to larger magnitudes ($U_c > 0$), eradicating the flat band in the non-interacting band structure. Nonetheless, the KB-QCP remains governed by a non-interacting FP. Throughout the OSM phase for $U > U_c$, we consistently observe the same universal behavior, consequently, our primary focus remains on the limit of zero $f$-bandwidth.

The effective SIAM at the critical point, $\alpha = 1$, corresponds to the well known and heavily studied pseudo-gap Anderson model [15–18]. As is customary for this model in the LM FP, both the effective medium, $\Im\Delta(z)$, and the imaginary part of the correlation self-energy, $\Im\Sigma(z)$, exhibit the same power-law exponent as the $f$-orbital spectra, albeit with opposite signs: $\Im\Delta(z) \propto |\omega|^{r^f}$, $\Im\Sigma(z) \propto |\omega|^{-r^f}$.

The phase space of the SIAM in the presence of a pseudo-gap DOS, characterized by an exponent $r < 0.5$, is well-documented for hosting an interacting non-Fermi-liquid FP with significant local moment fluctuations [15–18]. This non-Fermi-liquid FP exhibits a remarkable $\omega/T$ scaling behavior [16], consistent with experimental observations in various heavy fermion compounds at criticality [5–7].

In a PH pseudo-gap quantum impurity problem with a power-law hybridization function $\propto |\omega|^r$, a critical Kondo coupling strength $J_c(r)$ was identified for $r < 1/2$ [54] which governs the transition from a stable LM FP to a the stable SC FP above $J_c(r)$. Even though the pseudo-gap of the effective medium is described by an exponent $r^f = 0.33 < 1/2$, we were not able to find any stable FP other than the LM FP. This peculiar result can be attributed to the nature of the pseudo-gap arising as a consequence of self-consistency rather than being an inherent feature [55]. While studies of the pseudo-gap SIAM generally assume the power-law dependency to occur over the whole bandwidth of the conduction band DOS, we only obtain such behavior at low frequencies $\omega < \tilde{D}$ with the low-energy effective band width $\tilde{D} < D$. While the exponent $r^f = 0.33$ is fixed, the effective strength of hybridization $\tilde{V}^2$ in the small interval $I = [-\tilde{D} : -\tilde{D}]$,

$$\tilde{V}^2 = -\frac{1}{\pi} \int_{-\tilde{D}}^{\tilde{D}} \Im\Delta(z)d\omega\,, \tag{29}$$

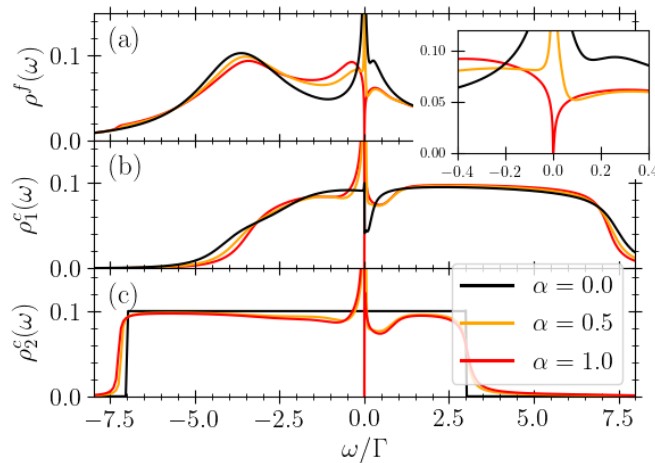

Figure 4: Same as Fig. 2 but for particle-hole asymmetric $f$-orbitals, $U/\Gamma = 10$, $\epsilon^{\mathrm{f}}/\Gamma = -3$.

depends on the DMFT self-consistency condition. While in a conventional pseudo-gap SIAM, the decrease of $U/\Gamma$ increases the Kondo coupling strength $J$ obtained by a Schrieffer-Wolff transformation [56], we observe a decrease in $\tilde{V}^2$ due to the DMFT self-consistency condition effectively stabilizing the LM FP.

However, even if the non-Fermi-liquid FP of the pseudo-gap Anderson model is absent in the effective site of the DMFT for the lattice model under investigation, we might be in its close proximity. As a result, at intermediate temperatures, various properties of the model might still exhibit hallmarks of the potentially nearby non-Fermi-liquid FP, including the $\omega/T$ scaling. This is similar to the apparent quantum-critical Mott transition in organic salts [57], which, at least from the perspective of one-band DMFT analysis, originates from approximate local quantum criticality [55].

### 4.3.3 PH asymmetric $f$-sites

We extend our analysis to PH $f$-sites, allowing for $\epsilon^{\mathrm{f}} \neq -U/2$. Figure 4 illustrates the spectral functions for the same parameters as in Fig. 2, but with $\epsilon^{\mathrm{f}}/\Gamma = -3$. The qualitative behavior of the spectra, when increasing $\alpha$, remains unchanged. There is a notable suppression of the Kondo resonance in the $f$-spectra, accompanied by the development of a peak in both $c$-spectra.

Remarkably, we again find a KB-QCP at $\alpha = 1$, supporting our conjecture that the QCP is driven by the absence of an conduction band induced inter-site hopping of the f-electrons in combination with an insufficient number of Kondo screening channels in the lattice. This KB-QCP, however, leads to modified low-frequency properties of the spectral functions which differ from the PH symmetric scenario.

Figure 5(a) displays the spectra of the $f$- and Fig. 5(b) $c_1$-orbitals at the KB-QCP on a low-frequency interval around the Fermi energy for four different values of $\epsilon^{\mathrm{f}}$ while keeping $U/\Gamma = 10$ fixed (both $c$-spectra are nearly identical on that scale). Importantly, not only the $f$- but also the $c$-spectra display a gap around $\omega = 0$, signaling an insulating ground state. The width of the gap is controlled by the degree of PH asymmetry and increases with the distance from the PH symmetric point. All spectra remain asymmetric down to zero frequency and display a discontinuity right at $\omega = 0$, formally indicating deviations from a Fermi-liquid description.

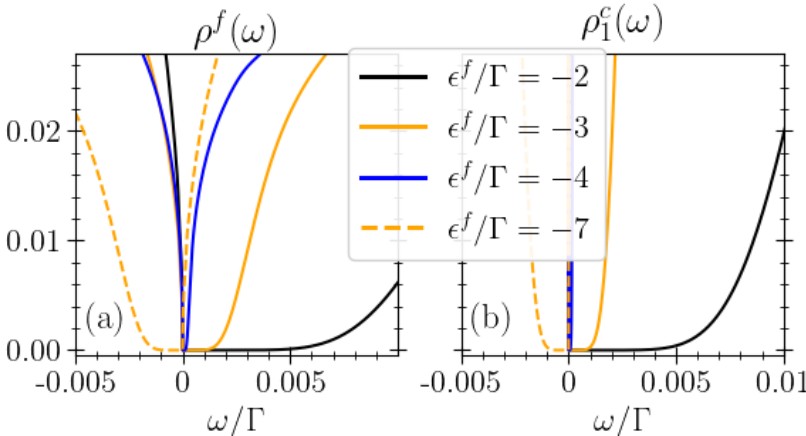

Figure 5: Small frequency regime of the single-particle spectral function of (a) the $f$-orbitals and (b) the $c_1$-orbitals at the KB-QCP, $\alpha = 1$, for locally particle-hole asymmetric $f$-orbitals. The different spectra correspond to $\epsilon^f/\Gamma = -2$ (black lines), $\epsilon^f/\Gamma = -3$ (orange lines), $\epsilon^f/\Gamma = -4$ (blue lines) and $\epsilon^f/\Gamma = -7$ (orange dashed lines), while $U/\Gamma = 10$ remains fixed. Other model parameters used are: $D/\Gamma = 5$, and $|\epsilon^c_\nu|/D = 0.4$.

For $\omega < 0$, the spectra are governed by an universal power-law dependency $\propto |\omega|^r$ with $r^f = 0.5$ and $r^c = -0.5$ independent of the degree of PH asymmetry or other model specifics like the shape of the conduction band DOS for less than half-filling. In contrast, for $\omega > 0$, the spectra vanish very quickly, and we were not able to extract the exponent of a possible power-law dependency. For larger than half-filling these universal exponents are found for $\omega > 0$.

It is essential to note that this insulating ground state differs from the usual Kondo insulator regime that typically occurs in the standard PAM around the PH symmetric point: (I) While local moments are screened in the Kondo insulator below a finite low-temperature crossover scale, $T_0 > 0$, here they remain unscreened down to zero temperature at the KB-QCP. Including non-local correlations by calculating two particle propagators [38, 58, 59] may lead to the occurrence of magnetic order or a spin liquid in the presence of frustration. (II) The width of the gap in the Kondo insulator is proportional to $T_0$ and, consequently, decreases with increasing strength of correlation, whereas the width of the gap in our case solely depends on the degree of particle-hole asymmetry. (III) As demonstrated in Ref. [36], in the Kondo insulator regime the effective site approaches the stable SC FP at low temperature which exhibits local Fermi-liquid properties. In the KB-QCP insulator, we obtain a discontinuity right at $\omega = 0$, violating such a description. In addition, the fixed point of the effective impurity is a local moment fixed point.

## 5 Summary and discussion

In this paper we studied a general multi-orbital extension of the widely recognized PAM within the framework of the DMFT+NRG. Focusing on the effect of two conduction bands with inverse dispersions to each other, hybridizing with a flat band of strongly correlated orbitals, we systematically analyzed the influence of the second band on the low energy Kondo scale $T_0$.

When the coupling to one of the bands dominates, we observe the emergence of well-

known characteristics of the standard PAM [36, 37]. This includes the presence of a Kondo resonance, with a width of $T_0$, right at the Fermi energy, and, depending on the degree of the PH asymmetry, an emerging depletion of the f-spectrum which develops into a full gap in the Kondo insulator regime. However, as the coupling to the second band is increased, when holding the overall hybridization matrix element to the f-orbital constant, the low-temperature Fermi-liquid scale $T_0$ decreases exponentially fast. It's important to note that adding arbitrarily shaped bands with non-conserved flavor to a SIAM always increases the local hybridization strength and the corresponding Kondo temperature, while this destructive interference and the associated drop of $T_0$ is a unique manifestation arising only in the lattice.

This exponential reduction of the Kondo lattice temperature $T_0$ can be explained by quantum interference of effective inter-site $f$-hopping elements [30] originating from the hybridization with different bands. This underlines the importance of these effective hopping elements over the single-site Kondo screening mechanism in the PAM [30].

Whereas the auxiliary two-band model enables a clean and systematic analysis of the effect of destructive hybridization interference, we showed that this will be a quite general feature in depleted versions of the standard PAM, relevant for a realistic description of heavy fermion materials where the unit cell typically comprise several atoms. In these cases the $f$-orbitals hybridize with multiple bands.

At high symmetry points with perfect destructive interference, the lattice low energy scale $T_0$ gets completely suppressed, leading to a Kondo breakdown critical point. In the DMFT self-consistency equations an effective pseudo-gap Anderson impurity model emerges which has a stable LM FP.

In the presence of locally PH symmetric $f$-orbitals, we observe power-law dependencies in all single-particle spectral functions, described by $A(\omega) \propto |\omega|^r$. Notably, the conduction band DOS remains metallic, with a diverging low-frequency spectral function characterized by an exponent of $r^c = -0.33$, while the correlated orbitals exhibit the development of a pseudo-gap with an exponent of $r^f = 0.33$. At this KB-QCP, the Fermi surface is reduced since it involves the conduction electrons only. The exponents do not depend on the the non-interacting metallic DOS of the conduction electrons or the strength of Coulomb interaction. Therefore, the critical interaction strength is zero, $U_c = 0$, however, a finite bandwidth for the correlated $f$-orbitals increases $U_c$ such that $U > U_c$ is required in order to reach this OSM phase.

Remarkably we find that the occurrence of the KB-QCP is neither associated with the PH symmetry or does it depend on the magnitude of the imaginary part of hybridization function as used in the traditional Doniach picture [25]. However, reducing the electron filling away from half filling, the KB-QCP is modified:

(I) A gap right at the Fermi energy is present in all spectral functions, demonstrating insulating nature of the solution. (II) For $\omega < 0$, we observe power law dependency in all spectra with modified universal exponents $r^f = 0.5$ and $r^c = -0.5$. (III) The size of the gap depends on the distance to the PH symmetric point, at which the gap turns into a true pseudo-gap. For increasing the electron filling above half filling, the universal exponents are observed for $\omega > 0$ as can be understood by a PH transformation.

We believe that our results are highly relevant for a broad range of strongly correlated materials. Versions of the PAM customized to specific heavy fermion materials will generally fall into the class of multi-orbital generalizations of the standard PAM. In such cases destructive hybridization interference can play a crucial role in understanding the low temperature behavior like the low-energy screening scale, determining the onset of universality.

Moreover, the notion of destructive hybridization interference provides a new avenue to explore Kondo breakdown scenarios within a rather general framework, without the need for intricate material-specific details. This avenue has the potential to enrich our existing comprehension of non-Fermi-liquid properties in strongly correlated materials.

# Acknowledgments

We acknowledge stimulating and fruitful discussions with B. Fauseweh and G. Camacho.

**Funding information** This project was made possible by the DLR Quantum Computing Initiative and the Federal Ministry for Economic Affairs and Climate Action; qci.dlr.de/projects/ALQU. F.B.A acknowledges funding by Deutsche Forschungsgemeinschaft through grant Nr. AN-275/10-1.

**Data availability** The data supporting the findings of this study are available from Zenodo at https://zenodo.org/records/10886260 and also upon request from the corresponding author.

# A DMFT integrals

In order to obtain results with high numerical precision it is very important to accurately solve the DMFT equation in Eq. (5). While the impurity Greens function in Eq. (7) are calculated within the NRG, the corresponding lattice Greens functions need to be calculated by evaluating Eq. (6). In case of the two-band model this equation reads

$$G_{\text{lat}}^{\text{f}}(z) = \int_{-\infty}^{\infty} \frac{\rho(\epsilon)\, d\epsilon}{z - \epsilon^{\text{f}} - \Sigma(z) - \frac{V_1^2}{z-(\epsilon+\epsilon^{\text{c}})} - \frac{V_2^2}{z+(\epsilon+\epsilon^{\text{c}})}} \tag{A.1}$$

$$= \int_{-\infty}^{\infty} \frac{\rho(\epsilon)((\epsilon+\epsilon^{\text{c}})^2 - z^2)}{F(z,\epsilon)}\, d\epsilon\,, \tag{A.2}$$

with

$$F(z,\epsilon) = a(z)\epsilon^2 + b(z)\epsilon + c(z)\,. \tag{A.3}$$

If we make use of the definitions $V^2 = V_1^2 + V_2^2$ and $\Delta V^2 = V_1^2 - V_2^2$, the factors $a(z)$, $b(z)$ and $c(z)$ are given by the following equations:

$$a(z) = z - \epsilon^{\text{f}} - \Sigma(z)\,, \tag{A.4}$$

$$b(z) = \Delta V^2 + 2\epsilon^{\text{c}} a(z)\,, \tag{A.5}$$

$$c(z) = z V^2 + \epsilon^{\text{c}} \Delta V^2 + ((\epsilon^{\text{c}})^2 - z^2) a(z)\,. \tag{A.6}$$

Now we can use partial fraction decomposition in order to rewrite $F^{-1}(z,\epsilon)$:

$$\frac{1}{F(z,\epsilon)} = \frac{A}{\epsilon - \epsilon_1} + \frac{B}{\epsilon - \epsilon_2}\,, \tag{A.7}$$

where the parameters are given as

$$\epsilon_1 = \frac{\sqrt{b(z)^2 - 4a(z)c(z)} - b(z)}{2a(z)}\,, \tag{A.8}$$

$$\epsilon_2 = \frac{-\sqrt{b(z)^2 - 4a(z)c(z)} - b(z)}{2a(z)}\,, \tag{A.9}$$

$$A = \frac{1}{a(z)(\epsilon_1 - \epsilon_2)}\,, \tag{A.10}$$

$$B = -A\,. \tag{A.11}$$

At this point the original DMFT integral can be written as

$$G(z) = \int_{-\infty}^{\infty} d\epsilon \left[ A\left( \frac{\rho(\epsilon)\epsilon^2}{\epsilon - \epsilon_1} - \frac{\rho(\epsilon)\epsilon^2}{\epsilon - \epsilon_2} \right) + 2\epsilon^c A\left( \frac{\rho(\epsilon)\epsilon}{\epsilon - \epsilon_1} - \frac{\rho(\epsilon)\epsilon}{\epsilon - \epsilon_2} \right) \right.$$
$$\left. + ((\epsilon^c)^2 - z^2)A\left( \frac{\rho(\epsilon)}{\epsilon - \epsilon_1} - \frac{\rho(\epsilon)}{\epsilon - \epsilon_2} \right) \right]. \tag{A.12}$$

If we now make use of the relations

$$\int_{-\infty}^{\infty} d\epsilon \frac{\rho(\epsilon)}{z - \epsilon} = \mathcal{G}(z), \tag{A.13}$$

$$\int_{-\infty}^{\infty} d\epsilon \frac{\rho(\epsilon)\epsilon}{z - \epsilon} = z\mathcal{G}(z) - 1, \tag{A.14}$$

$$\int_{-\infty}^{\infty} d\epsilon \frac{\rho(\epsilon)\epsilon^2}{z - \epsilon} = z(z\mathcal{G}(z) - 1), \tag{A.15}$$

which hold for a particle hole symmetric DOS $\rho(\epsilon) = \rho(-\epsilon)$ we finally obtain

$$G(z) = A\left[ \left( \epsilon_1 - \epsilon_1^2 \mathcal{G}(\epsilon_1) \right) - \left( \epsilon_2 - \epsilon_2^2 \mathcal{G}(\epsilon_2) \right) \right] + 2\epsilon^c A\left[ (1 - \epsilon_1 \mathcal{G}(\epsilon_1)) - (1 - \epsilon_2 \mathcal{G}(\epsilon_2)) \right]$$
$$+ (z^2 - (\epsilon^c)^2)A\left[ \mathcal{G}(\epsilon_1) - \mathcal{G}(\epsilon_2) \right]. \tag{A.16}$$

The Hilbert transformation in Eq. (A.13) can be solved with very high accuracy. For example, in case of a semicircular DOS the analytical solution is known. The DMFT integrals for the $c$-orbital spectra can be calculated in a similar way.

## B  Multi-band SIAM vs. multi-band PAM

Here we shortly discuss the difference of having multiple bands in the single impurity limit and the corresponding lattice model.

Assuming that the impurity is coupled with identical strength $V$ to multiple bands, the SIAM hybridization function reads:

$$\Delta^{\text{SIAM}}(z) = V^2 \sum_{\vec{k}} \sum_{\nu} (z - \epsilon_{\vec{k}}^{\nu})^{-1}, \tag{B.1}$$

where $\nu$ indicates the band indices. The influence of the conduction bands onto a single-orbital SIAM under consideration here is fully determined by the hybridization function $\Delta^{\text{SIAM}}(z)$ independent of the number of bands. Furthermore, the momentum vector $\vec{k}$ and band index $\nu$ contribute in exactly the same way, such that we can also define a combined index $\vec{g} = (\vec{k}^T, \nu)$ and a new dispersion $\epsilon_{\vec{g}} = \epsilon_{\vec{k}}^{\nu}$ to rewrite Eq. (B.1) in the form of the common single-band SIAM. Consequently, in the case of a single-orbital SIAM coupled to multiple bands, there is always an equivalent single-band SIAM.

One might think that this also holds true in the case of the multi-band PAM treated with DMFT, as the lattice problem is mapped onto an effective SIAM. However, this is not the case. The reason is obvious: While the spatial information is lost in a SIAM as indicated in Eq. (B.1), the lattice problem is diagonalized in $\vec{k}$-space to accommodate the translation invariance of the problem. The lattice symmetries and multi-orbitals of the conduction bands are encoded in $\Delta_{\vec{k}}^{\text{lat}}(z)$, which turns out to be of crucial importance for the spectral properties and marks the difference between SIAM and a DMFT problem.

Within each DMFT iteration, the effective medium $\Delta^{\text{eff}}(z)$ of the SIAM is calculated by Eq. (9), where $G^{\text{f}}_{\text{lat}}(z)$ contains the relevant information about the lattice model under investigation. From Eqs. (6) and (8), one can see that momentum $\vec{k}$ and band index $\nu$ contribute in a different way: whereas the summation over $\vec{k}$ is in the numerator, the summation over $\nu$ is in the denominator of $G^{\text{f}}_{\text{lat}}(z)$, such that it's not possible to combine both indices as simply as $\vec{g} = (\vec{k}^T, \nu)$.

Instead, if we want to map the multi-band PAM onto an equivalent single-band PAM, with dispersion $\tilde{\epsilon}_{\vec{k}}$ and coupling $\tilde{V}$, we need to ensure that $\Delta^{\text{lat}}_{\vec{k}}(z)$ remains invariant as well:

$$\Delta^{\text{lat}}_{\vec{k}}(z) = \sum_{\nu} V_{\nu}^2 (z - \epsilon_{\vec{k}})^{-1} \stackrel{!}{=} \tilde{V}^2 (z - \tilde{\epsilon}_{\vec{k}})^{-1} . \tag{B.2}$$

If all the different bands are exactly identical, this equation is trivially solved by $\tilde{\epsilon}_{\vec{k}} = \epsilon_{\vec{k}}$ and $\tilde{V}^2 = \sum_{\nu} V_{\nu}^2$. However, if the bands have a different dispersion, $\tilde{\epsilon}_{\vec{k}}$ and $\tilde{V}^2$ obtained from Eq. (B.2) will, in general, depend on frequency, rendering it uninterpretable as non-interacting band and coupling, respectively.

To conclude, whereas it is always possible to map the multi-band SIAM onto an equivalent single-band problem, this is, in general, not possible in the case of the PAM.

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
