# Peer review of "Kondo breakdown in multi-orbital Anderson lattices induced by destructive hybridization interference"

_SciPost Physics, doi:SciPost Phys. 17, 069 (2024)_

## Round 1 · Referee Report · Anonymous (Referee 1) · 2024-3-1

Weaknesses

1- The claim of generality and relevance appears too strong. In my opinion, the results are not as generally valid and I am not quite convinced that the symmetry relation for the two dispersion relations is expected to be commonly satisfied.

2- I also find it regrettable that the authors do not provide any data sets nor scripts/codes for calculations and postprocessing of the results: this makes it difficult to independently verify the numeric results, especially to non-practitioners of the methods used.

Report

This work presents a scenario which leads to a Kondo breakdown mechanism in a system with a particular dispersion relation for conductance electrons in a two-orbital periodic Anderson model. The effect is shown to be due to interference. The spectrum is found to exhibit power-law scaling and a peculiar strong asymmetry in the case of particle-hole asymmetry of the Hamiltonian. I find the reported observations interesting because the work presents a novel and rather unexpected mechanism that can lead to a Mott phase. In particular, I like the explanation about the particular lattice effects and the role of the real part of the hybridisation self-energy.

The paper is clearly written, it appears technically correct, as far as I can tell, and it presents a good level of detail. The results are properly interpreted. I recommend this work for publication. I would, however, recommend that the claim of generality and relevance be toned down.

I have some questions:

  • Is there some understanding about the origin of the exponent value 0.33? Can one relate this problem to pseudo-gap Anderson impurity model physics and known relation between exponents (Glossop, Logan (2000))?

  • Do the dispersion need to be "inverse" only asymptotically or throughout the band?

There are some typesetting issues and at least one typo:

  • Missing vector sign in k in Eq. (2)
  • Sum symbol mu in Eq. (4) should probably be nu
  • "quit general" -> "quite general"
  • validity: high
  • significance: good
  • originality: high
  • clarity: top
  • formatting: good
  • grammar: excellent

Author:  Fabian Eickhoff  on 2024-03-08  [id 4351]

(in reply to Report 1 on 2024-03-01)

We appreciate the referee's overall positive feedback and insightful comments, which we will thoroughly address in a possible resubmission.

Regarding the claim of generality and relevance: The primary message conveyed in the paper is that in lattice models, the screening of local moments is predominantly influenced by the real part of the hybridization function. This stands in stark contrast to the single impurity limit, where the low-energy properties are primarily determined by the imaginary part. Therefore we address two key questions: (I) We analyze the possibility of destructive hybridization interference and its accompanying exponential suppression of the low-energy screening scale, $T_0$, in lattice models. This effect is quite general and occurs whenever contributions to the real part of the hybridization function in Eq. (10) differ in sign. For this phenomenon to manifest, the dispersions do not need to be exactly inverse but simply need to exhibit different curvatures near the Fermi energy. This scenario corresponds to $V_1 \neq V_2$ as described in the paper but can also be realized by setting $\epsilon_{k 2} = -\epsilon_{k 1} + \delta\cdot\gamma_k$, with $\gamma_k$ being an arbitrary perturbation. As analytically demonstrated in Sec. 4.2, this destructive interference will be relevant in more realistic depleted generalizations of the standard PAM. (II) Having established the significant influence of destructive interference on $T_0$, we consider the extreme case of the Kondo breakdown critical point where $T_0=0$. As stated in the abstract, this occurs "At high symmetry points" only. We agree with the referee that this critical point requires fine-tuning and is not a general feature of the extended PAM.

Regarding the understanding of the exponent value 0.33: Upon achieving self-consistency within the DMFT framework, well-known results for the pseudo-gap Anderson model can be applied, as demonstrated in Sec. 4.3.2 on page 13. However, the exponent value of 0.33 is generated through the self-consistency cycle. It may be an interesting avenue for future research to investigate whether analytical insight can be gained from the DMFT equations regarding this exponent value.

Regarding the asymptotics of the band dispersion: As mentioned in our response to the first comment, the dispersions do not need to be "inverse" for the occurrence of destructive interference and an exponentially suppressed $T_0>0$. The question regarding the stability of the Kondo breakdown critical point with respect to deformations of the two bands is indeed crucial. Indeed, currently we are actively investigating this aspect and hope to provide a comprehensive answer to this pertinent question in our next publication.

Anonymous on 2024-04-12  [id 4413]

(in reply to Fabian Eickhoff on 2024-03-08 [id 4351])
Category:
remark

The response is satisfactory for the most part. I do believe, however, that saying "predominantly influenced" (by the real part) is a rather strong assertion. The real and imaginary part of the hybridisation function of the effective impurity level are anyhow connected via the Kramers-Kronig relations, there is only an overall shift in the real part that is indeterminate at this level.
In spite of this, I appreciate the analysis performed in this work about what determines the hybridisation (including the said overall shift).

---

## Round 1 · Referee Report · Anonymous (Referee 2) · 2024-3-12

Report

This paper introduces a new mechanism producing Kondo breakdown in multi-band versions of the periodic Anderson model. The effect originates from a destructive interference of the bands occurring from canceling contributions produced by effective intersite hoppings.
The QCP occurs at a critical point ($\alpha_c=1$) of an effective parameter $\alpha$ describing the exact destructive interference and consequently a vanishing of the corresponding Kondo temperature.
This fact is very pedagogically explained in this paper by starting from a toy model with two bands with opposite curvature. The author subsequently demonstrate that the effect is generic to PAMs with multiple conduction bands.
In section 4.3 the authors study the critical properties of this transition, identify the exponential suppression of the energy scale with $\alpha -1$ and interestingly identify an universal power law of the spectrum at the QCP. They discuss the relation, but in particular the differences with the properties of the pseudogap AM.
Also in the particle-hole asymmetric case a QCP is identified again with power-law behavior of the spectra.
Identifying a new kind of QCP in a heavy fermion system is certainly a very interesting result. These properties might be explored experimentally in specific materials. In addition, the paper is written very clearly and pedagogically, combining semi-analytical description for a general understanding with detailed solid numerical analysis. For this reason, I clearly recommend this work for publication in SciPost.

Requested changes

1) Eqs. 2,3,4: $\mu$ in the sums should be $\nu$ 2) Eq. 4: $c_l$ is clearly the FT of $c_k$ but this is not written anywhere 3) Eq 4: In the second sum, l goes up to $N_f$. For this reason, it may be useful to name $N_f$ as $N$ and denote it as the "number of lattice sites". Maybe I would also rename $N_\nu \rightarrow n_\nu$ (lowercase) since this is not an extensive quantity 4) Eq. 9 $i \Im{}m \Delta_{lm}$ ($\Im{m}$ is missing) and a question here, is it the real/imaginary part or rather the hermitian/antihermitian parts of the matrix $\Delta$ ? 5) Above Eq. 17 : $\tilde\Gamma$ I guess it's the overall scale. I didn't find a definition for it though 6) After having define $\gamma$ in terms of $\alpha$, in subsequent discussions please choose either $\alpha$ or $\gamma$ to describe the properies approaching the QCP, since it's equivalent. Otherwise at some point you write $\gamma\to\infty$, later $\alpha\to1$ and the reader has to remember that it's the same thing. For example at the beginning of Sec. 4.3 the sentence ".. $\alpha=\cdots =1$, and $\gamma=\infty$ .." seems to indicate that these are two independent conditions that have to be realized 7)Sec. 5, 4 pargraph quit->quite

  • validity: high
  • significance: high
  • originality: high
  • clarity: top
  • formatting: excellent
  • grammar: perfect

Author:  Fabian Eickhoff  on 2024-03-18  [id 4373]

(in reply to Report 2 on 2024-03-12)

Dear Referee,

Thank you for your positive feedback and recommendation for publication in SciPost.
We will address all your remarks in our resubmission to ensure the clarity and rigor of our research.

---

## Round 1 · Referee Report · Hugo Strand (Referee 3) · 2024-3-13

Report

The authors study the periodic Anderson model within the dynamical mean-field theory (DMFT) approximation using a numerical renormalization group solver for the effective impurity problem. Within DMFT the solution is only sensitive to the density of states of the hybridization function of the itinerant electron species, which the authors model with a weighted sum of two square density of states with equal bandwidth, symmetrically shifted above and below the Fermi level. The spectral features are then studied as a function of the weight parameter.

I find the numerical results convincing and the concept of tuneability using two species of itinerant electrons is interesting. However, I do not share the authors conviction that the physics is fundamentally different from the periodic Anderson model with a single itinerant electron species, in particular in the dynamical mean-field approximation applied.

I think that an identical spectral result for the f-electron spectral function (Figs. 2 - 5) can be obtained from the single itinerant band periodic Anderson model, where the single itinerant species is having a density of states comprised of a weighted sum of two square functions. At the special point (\alpha = 1, in the manuscript notation) this DOS is particle hole symmetric and its Hilbert transform (giving the real part of the hybridization) is odd in frequency and zero at frequency zero. Hence, the system is back to the Kondo insulating regime of the periodic Anderson model with a single itinerant species.

Before the authors convincingly show that the setup with two itinerant bands is fundamentally different from a single itinerant band model (with the same itinerant electron density of states) I can not recommend the manuscript for publication in SciPost Physics.

Requested changes

  • Provide reproducibility-enabling resources

I recommend the authors to amend the manuscript with a repository with simulation scripts, numerical results, and visualization scripts. In order to make the numerical results available and reproducible to the community.

  • Reference to NRG implementation in the TRIQS package

In the manuscript the authors state:

"... we employ the Numerical Renormalization Group (NRG) technique, as implemented in the TRIQS open-source package [42]."

As a TRIQS developer I know that there is no open source NRG solver in the TRIQS package.

Reference 42 pertains to the NRG code by Rok Žitko called "NRG Ljubljana" (http://nrgljubljana.ijs.si/) whose landing page recommends citations of PRB 79, 085106 (2009) and https://doi.org/10.5281/zenodo.4841076.

The TRIQS package does have an interface to the NRG Ljubljana code https://triqs.github.io/nrgljubljana_interface and academic use of this package is recommended to be combined with a citation of the article:

O. Parcollet, M. Ferrero, T. Ayral, H. Hafermann, I. Krivenko, L. Messio, and P. Seth, Comp. Phys. Comm. 196, 398-415 (2015)

I recommend the authors to cite all three DOIs.

  • Abstract: Undefined quantities in the equations

The abstract is not self contained since the two temperature scales presented contain symbols (\alpha, U, \Gamma_1, \Gamma_2) that are not defined there. I recommend the authors to make the abstract self contained with regards to symbolic notation.

  • Eq. 10 missing additional f-orbital index?

In the text before Eq. (10) the authors generalize the formalism to multiple f- and c- electrons per unit cell. However, in Eq. (10) only multiple c- electron indices appear. I think removing the statement of multiple f- electrons per unit cell would harmonize the text and equations.

  • Explicitly define \Gamma, \tilde{\Gamma}, and T_0

The authors use notation from their previous works that is not clearly defined in the manuscript. Please relate these quantities to preceding model quantities to help guide the reader.

  • Abbreviation: fix point (FP)

Please consider writing fix point out verbatim. I think the multiple combined abbreviations is reducing the readability of the manuscript.

  • validity: -
  • significance: -
  • originality: -
  • clarity: high
  • formatting: excellent
  • grammar: perfect

Author:  Fabian Eickhoff  on 2024-03-18  [id 4372]

(in reply to Report 3 by Hugo Strand on 2024-03-13)

Dear Mr. Hugo Strand,

Thank you for your thoughtful assessment of our manuscript and for providing constructive feedback, particularly by providing the relevant references concerning the TRIQS-NRG framework. We appreciate the opportunity to address your concerns and clarify the significance of our findings:

  • "Before the authors convincingly show that the setup with two itinerant bands is fundamentally different from a single itinerant band model (with the same itinerant electron density of states) I can not recommend the manuscript for publication in SciPost Physics" The primary critique revolves around the assertion that the physics described in our two-band model can be replicated within an effective single-band framework. However, as we demonstrate below, this assumption oversimplifies the complexities inherent in our study. Firstly, given the lattice nature of our problem, the hybridization function depends on momentum. In the case of our two-band model, the hybridization function is given by Eq. (14) in our paper. to map this problem onto an effective single band model with effective dispersion $\tilde{\epsilon}^c_k$ and coupling $\tilde{V}$, such that the DMFT self-consistency equations remain invariant, we need to solve $\frac{V_1^2}{z-\epsilon^c_{k1}}+\frac{V_2^2}{z-\epsilon^c_{k2}} \stackrel{!}{=}\frac{\tilde{V}^2}{z-\tilde{\epsilon}^c_{k}}$ for $\tilde{\epsilon}^c_k$. While for $\epsilon^c_{k1}=+\epsilon^c_{k2}$, the solution trivially yields $\tilde{\epsilon}^c_k=\epsilon^c_{k1}=\epsilon^c_{k2}$, in general, when $\epsilon^c_{k1}\neq\epsilon^c_{k2}$, $\tilde{\epsilon}^c_k$ becomes frequency-dependent, rendering it non-interpretable as a non-interacting band.

  • " At the special point (\alpha = 1, in the manuscript notation) this DOS is particle hole symmetric and its Hilbert transform (giving the real part of the hybridization) is odd in frequency and zero at frequency zero. Hence, the system is back to the Kondo insulating regime of the periodic Anderson model with a single itinerant species." The referee mixes up a local DOS of a single conduction band and the resultung effecive DOS of the media entering the DMFT self-consistency condition. The assertion that the real part of the hybridization function vanishes under particle-hole symmetry holds true only for an impurity problem. In the lattice scenario, characterized by a single band $\epsilon_k$, the hybridization function $\Delta_k(z) = \frac{V^2}{z-\epsilon_{k}}$ has non-zero real part at zero frequency, unless the dispersion is divergent. Therefore, to achieve a scenario where $\Re\Delta_k(0+i\delta)$ vanishes for all momenta $k$, a second band is necessitated. The Kondo insulator has a divergent part in the imaginary part of the media as it is well known (Pruschke et al PRBPhys. Rev. B 61, 12799 (2000). Pruschke et al pointed out, that this delta peak is treated by coupling a zero energy orbital to the Anderson impurity in the DMFT self-consistency. The physical origin of this term is the effective f-f coupling due to \Real \Delta_{ij}\not = 0 and is responsible for the screening of the local moments in the Kondo insulator. We, however, present a Kondo Breakdown point with unscreened moments which we illustrate by mapping the problem onto a depleted PAM where a finite large moment is well established by the Lieb-Mattis theorem. We will summarize the qualtativ differences between the Referee's Kondo-insulator scenario and our Kondo breakdown point below:

    • Local moments in the Kondo insulator are screened below a finite low-temperature crossover scale, while they remain unscreened at the KB-QCP, even at zero temperature.
    • The Kondo insulator converges to the stable strong-coupling fixed point, whereas the KB-QCP converges to the local moment fixed point.
    • In the Kondo insulator, c-orbitals exhibit a gapped density of states, unlike the metallic behavior observed in the KB scenario with particle-hole symmetric f-sites.
    • The Kondo insulator smoothly connects to the non-interacting limit, whereas at the KB-QCP, the imaginary part of the self-energy diverges, $\Im\Sigma(\omega+i\delta)\propto |\omega|^{-1/3}$, indicative of a Mott-insulating f-site nature.

Anonymous on 2024-04-12  [id 4414]

(in reply to Fabian Eickhoff on 2024-03-18 [id 4372])
Category:
remark

In the response to the question regarding the difference between single and two-band descriptions, I don't understand what is meant by "the hybridisation function depends on momentum". At the end of the day, what is relevant for the effective impurity problem is only the frequency-dependent (complex valued) hybridisation function, irrespective of how the different k-modes contribute to it. I don't see how the contributions of a particular k1 and a particular k2 modes are relevant.
At the technical level, by allowing energy dependent couplings V, one can map any DOS to a single non-interacting band.
The point is rather that such a mapping is not realistic. On the other hand, with two bands, one can obtain the DOS of interest in a system that is physically realistic, which is what this work demonstrates.

Anonymous on 2024-04-12  [id 4415]

(in reply to Anonymous Comment on 2024-04-12 [id 4414])

Thank you very much for your comment.
As this point seems to be crucial for the discussion we will try to convince you with adding some more details about SIAM and DMFT:

In case of the single impurity problem the hybridization function depends on frequency only,
and one can always map a setting where multiple bands are coupled to the impurity to an effective single band model.
The hybridization function $\Delta_\text{SIAM}(z)$ of the SIAM with multiple bands reads:
$\Delta_\text{SIAM}(z)=\sum_{\vec{k}}\sum_i \frac{V^2}{z-\epsilon^i_{\vec{k}}}=\sum_\gamma \frac{V^2}{z-\tilde{\epsilon}_\gamma}$.
Here one can clearly see that multiple bands do not have any effect as the band index $i$ and the momentum vector $\vec{k}$ are treated identically, such that one can also create a new combined index $\gamma$.

However, in our paper we are NOT studying a single impurity problem but treat a multi band lattice model with DMFT.

Within DMFT the lattice problem is mapped onto an effective SIAM, characterized by the effective hybridization function, $\Delta_\text{DMFT}(z)$, which only depends on frequency.
The self-consistency equation reads:
$\Delta_\text{DMFT}(z)=z-\epsilon^f-\Sigma(z)-G^{-1}_\text{loc}(z)$.
$G_\text{loc}(z)$ is the local Greens function of the lattice model one is interested in. This Greens function exhibits the physical relevant hybridization function $\tilde{\Delta}_{\vec{k}}(z)$ which includes information about the lattice model under investigation.
$G_\text{loc}(z) = \sum_{\vec{k}} \frac{1}{ z-\Sigma(z) - \tilde{\Delta}_{\vec{k}}(z)}$
$\tilde{\Delta}_{\vec{k}}(z) = V^2\sum_i (z-\epsilon^i_{\vec{k}})^{-1}$
In this case one can NOT swap $\sum_i$ and $\sum_k$ or create a new combined index, as $\sum_k$ appears in the nominator and $\sum_i$ in the denominator of $G_\text{loc}(z) $.

Consequently, if one still wants to map the multi band lattice model onto a single band lattice model (in which one also alowes for energy dependent coupling), such that the DMFT equations or $G_\text{loc}(z)$ respectively remains invariant , one needs to ensure that $\tilde{\Delta}_{\vec{k}}(z)$ remains invariant:
$\sum_i \frac{V^2}{z-\epsilon^i_{\vec{k}}} \stackrel{!}{=} \frac{\tilde{V}^2_{\vec{k}}}{z-\tilde{\epsilon}_{\vec{k}}}$
In general there is no solution with frequency independent $\tilde{\epsilon}_{\vec{k}}$ and $\tilde{V}^2_{\vec{k}}$, rendering it uninterpretable as a single band model.

In conclusion, one can always map a multi band SIAM onto an effective single band SIAM, whereas in general one can not map a multi band PAM onto an effective single band PAM.

---

## Round 3 · Referee Report · Hugo Strand (Referee 3) · 2024-5-7

Strengths

  • Novel idea of tuneability from interference between multiple non-interacting itinerant bands in the periodic Anderson model.

  • The numerical results are available from a separate Zenodo repository.

Weaknesses

  • As I and one other referee pointed out in our first reports the manuscript does not adhere to the SciPost Physics author guidelines on the point

"""Supplementary Material: You should make supplementary material (data, experimental details, analyses, code and similar) openly available, most appropriately by depositing those in institutional or public repositories. In your paper, please list the items together with the link pointing to where they are hosted."""

See https://scipost.org/SciPostPhys/authoring

In the resubmission the authors have supplied the numerical results producing the figures in the manuscript. However, the means of reproducing the results have not been made available, even though the calculations have been performed using publicly available open source software.

Report

In the revised manuscript the authors have made a serious effort to address the issues raised by the referees. Unfortunately, I think the readability of the manuscript has suffered in this effort since my main point of critique was based on a misunderstanding from my side, see details below.

The current manuscript contains several sections that I think could be delegated to appendices, in favor of a more concise story in the main text. I recommend the authors to consider making appendices of the sections, "2.2.1 Multi-band SIAM vs. Multi-band PAM", "3 Theoretical motivation", and "4.2 Generality of destructive hybridization interference".

All in all, I think that the results and manuscript text now meet the criteria for publication in SciPost Physics. However, I do not think the manuscript meets the SciPost Physics criteria of reproducibility, since the NRG simulation scripts are not part of the supplemental materials distributed on Zenodo.

If the authors make the results reproducible by extending the supplemental Zenodo material and considers the comments in my report I am willing to support the manuscript for publication in SciPost Physics.

Two-band PAM and interference effects

In my first report I raised the question how the two-band PAM differs from the single band PAM and asked the authors to clarify. The authors reply and the ensuing discussion with another referee was enlightening on this point and I now think I understand the fundamental difference between the two models. However, in my opinion, the revised manuscript remains obscure on this point.

I think the introductory sections on the two-band PAM could benefit from being more concrete and explicit. In particular I recommend the authors to explicitly include the first row of Eq. (A.1) in the method section, since this is explicitly how the authors perform the integration over lattice momenta. With this equation it is possible to realize that there are no functions $\rho_1(\epsilon)$ and $\rho_2(\epsilon)$ that can satisfy the relation

$$ G^f(z) = \int_{-\infty}^\infty d\epsilon \, \rho^c(\epsilon) \frac{1}{ z - \epsilon^f - \Sigma(z) - \frac{V_1^2}{z - (\epsilon + \epsilon^c)} - \frac{V_2^2}{z - (\epsilon + \epsilon^c)} } \ \ne \int_{-\infty}^\infty d\epsilon \, \rho_1(\epsilon) \frac{1}{ z - \epsilon^f - \Sigma(z) - \frac{V_1^2}{z - (\epsilon + \epsilon^c)} }
\ + \int_{-\infty}^\infty d\epsilon \, \rho_2(\epsilon) \frac{1}{ z - \epsilon^f - \Sigma(z) - \frac{V_2^2}{z - (\epsilon + \epsilon^c)} }
$$
which was the cause of confusion in my first report.

Further I think that the discussion on the small energy scale produced by the interference between the two bands could be made more informative by explicitly showing the non-interacting f-electron density of states $\rho^f(\omega)$ and how a low energy scale appears as $\alpha \rightarrow 1$ already in the non-interacting limit.

It would also be instructive to compare $\alpha=1$ and $U=0$ case for $\rho^f(\omega)$ with the single band PAM density of states from

$$ G^{f}{SB-PAM}(z) = \int^\infty d\epsilon \, \rho^c(\epsilon) \frac{1}{ z - \epsilon^f - \Sigma(z) - \frac{V^2}{z - \epsilon} } $$
and seeing that, while both are gapped, the two-band PAM has an additional sharp delta-peak resonance at $\alpha = 1$ located at $\omega = 0$.

Claim: Scaling of low energy scale $T_0$.

With regards to the data shown in Fig. 1 the authors state

"""For small values of γ − 1, we observe an excellent agreement between the ratio determined numerically with the DMFT(NRG) and analytic expression stated in Eq. (21).""

In this regime the value of the ratio is of order unity (1) while the deviations of the numerical data from Eq. (21) is of the order 1/10. The log-scale on the y-axis obfuscates this somewhat.

I recommend the authors to weaken their claim that a 10% absolute error is an excellent agreement.

Claim: Quantum phase transition at $\alpha = 1$

Regarding the quantum phase transition and disappearance of the low energy temperature scale $T_0$ as $\alpha \rightarrow 1$, the current presentation in the manuscript connects this to an effective Kondo lattice scale.

However, as far as I can tell, the two-band PAM undergoes a quantum phase transition as $\alpha \rightarrow 1$ already in the non-interacting ($U=0$) limit. As $\alpha \rightarrow 1$ (when $U=0$) the gap in the f-electron spectral density goes to zero, and at $\alpha = 1$ the gap closes and the system becomes metallic with a sharp resonance at $\omega = 0$.

I recommend the authors to clarify the origin of the quantum phase transition in the two-band PAM and how the phase transition at finite $U$ is connected to the "trivial" phase transition on the non-interacting limit.

Requested changes

  • Delegate some of the discussions in the introduction to appendices.

  • Explicitly state used lattice self consistency relation in the introduction, see 1st line in Eq. (A.1)

  • Fig. 1: Weaken statement on "excellent agreement" given 10% absolute errors at small $\gamma - 1$.

  • Plot $\rho^f(\omega)$ for $U = 0$, $|\epsilon^{c}|\ne 0$ and $\alpha = 0, 1/2, 3/4, 1, 4/3, 2, \infty$ showing how the band interference effects change with $\alpha$ and induce a low energy scale that goes to zero as $\alpha = 1$.

  • Make numerical simulations reproducible by distributing the python scripts operating Ljubjana NRG on Zenodo.

  • Clarify quantum phase transition claim

  • Fig. 2: The color descriptions in the figure caption does not agree with the figure legend.

Recommendation

Ask for minor revision

---

## Round 3 · Author Response

We extend our sincere gratitude to all the referees for their thoughtful and constructive comments and recommendations. Based on their feedback, we have implemented several modifications to enhance the quality and clarity of our manuscript.

A significant alteration involves providing a more comprehensive description of the Dynamical Mean Field Theory method. We have incorporated a new subsection dedicated to delineating the distinctions between a multi-band single impurity model and a multi-band lattice model. This addition aims to elucidate the intricacies of our approach, thereby enriching the understanding of readers and reviewers alike.
In addition we also created a data repository and published the data used in our manuscript.

We trust that these revisions effectively address the concerns raised by the referees and underscore the importance and significance of our analysis.

---

## Round 3 · List of Changes

• added more details in 'Method' section
  • added new subsection 'Multi-band SIAM vs. Multi-band PAM'
  • added paragraph Data availability
  • added "NRG-Lubljana interface" and citations [43] and [44] in the TRIQS context
  • added text block in Sec. 4.1 starting with "Please note that ..."
  • modified abstract and added definitions of $U$ and $\Gamma_i$ therein
  • changed $k$ to $\vec{k}$ in Eq.(2)
  • changed $\mu$ to $\nu$ in Eq.(2) and Eq.(4)
  • changed $i$ to $\nu$ in Eq.(16)
  • changed "quit" to "quite"
  • added definition of $c_{l\sigma}$
  • added $\Im$ in Eq.(9)
  • changed $\gamma$ to $\gamma(\alpha)$

---

## Round 4 · Author Response

We again thank the referees for their valuable feedback, which has helped us to further improve the quality of our manuscript. We have addressed the reviewers' comments and made several revisions accordingly. However, these changes have not altered the substantive content of the manuscript. We believe these changes have strengthened the readability of the manuscript and addressed the reviewers' concerns comprehensively.

---

## Round 4 · List of Changes

- add reference to appendix A and Eq. A1 in the Method section where we explain the DMFT algorithm
- moved subsection 'Multi-band SIAM vs. Multi-band PAM' to appendix B.
- replaced 'excellent agreement' by 'qualitatively good agreement' in the context of Fig.1
- fixed color descriptions in the figure caption of Fig.2
- Extend discussion in Sec. 4.3.2 in the context of the emerging flat band in the non-interacting limit starting with "It's crucial to emphasize..."

---

## Editorial Decision

published